

# Carroll fermions

**Eric A. Bergshoeff[1⋆], Andrea Campoleoni[2†], Andrea Fontanella[3,4‡],
Lea Mele[2∘] and Jan Rosseel[5§]**

**1** Van Swinderen Institute, University of Groningen,
Nijenborgh 4, 9747 AG Groningen, The Netherlands
**2** Service de Physique de l'Univers, Champs et Gravitation, Université de Mons – UMONS,
20 place du Parc, 7000 Mons, Belgium
**3** Perimeter Institute for Theoretical Physics, Waterloo, Ontario, N2L 2Y5, Canada
**4** School of Mathematics & Hamilton Mathematics Institute, Trinity College Dublin, Ireland
**5** Division of Theoretical Physics, Rudjer Bošković Institute,
Bijenička 54, 10000 Zagreb, Croatia

⋆ e.a.bergshoeff@rug.nl , † andrea.campoleoni@umons.ac.be , ‡ andrea.fontanella@tcd.ie ,
∘ lea.mele@umons.ac.be , § Jan.Rosseel@irb.hr

## Abstract

Using carefully chosen projections, we consider different Carroll limits of relativistic Dirac fermions in any spacetime dimensions. These limits define Carroll fermions of two types: electric and magnetic. The latter type transforms as a reducible but indecomposable representation of the Carroll group. We also build action principles for all Carroll fermions we introduce; in particular, in even dimensions we provide an action principle for a minimal magnetic Carroll fermion, having the same number of components as a Dirac spinor. We then explore the coupling of these fermions to magnetic Carroll gravity in both its first-order and second-order formulations.

# 1 Introduction

The Carroll group refers to the particular Inönü-Wigner contraction of the Poincaré group that is physically interpreted as its ultra-relativistic limit, in which the speed of light $c$ is taken to 0. Originally studied by Lévy-Leblond [1] and Gupta [2], Carroll symmetries have recently received attention in the context of flat space holography (see, e.g., the recent reviews [3, 4]) and in investigations of the physics of black hole horizons [5]. This renewed interest stems from the fact that the conformal extension of the Carroll group is the so-called BMS group that describes the asymptotic symmetries of flat spacetime at null infinity [6] and that any null hypersurface is described by a manifold whose structure group is the Carroll group. Carroll symmetries also naturally arise in the tensionless limit of string theory; see, e.g., [7, 8] and references therein.

Motivated by these applications of Carroll symmetry, various authors have recently considered the construction and study of Carroll invariant field theories [9–26]. Such field theories are usually obtained by taking a $c \to 0$ limit of relativistic ones and this can typically be done in two different ways. Correspondingly, two different types of Carroll field theories, often dubbed 'electric' and 'magnetic', are obtained in the limit. To illustrate these electric and magnetic Carroll limits, let us follow [12,13] and consider a relativistic free real scalar $\Phi$ with mass $M$, whose Lagrangian is given in Hamiltonian form by

$$\mathcal{L} = \Pi_\Phi \partial_t \Phi - \frac{c^2}{2} \Pi_\Phi^2 - \frac{1}{2} \partial_a \Phi \partial^a \Phi \mp \frac{(Mc)^2}{2} \Phi^2, \tag{1}$$

where $\partial_a$ denotes the derivatives with respect to the spatial coordinates and $\Pi_\Phi$ is the momentum conjugate to $\Phi$. We have furthermore reinstated the speed of light $c$, by setting the time-like coordinate equal to $ct$, and left the sign in front of the mass term arbitrary, so that we allow the scalar to be tachyonic. The Lagrangian of the real free electric Carroll scalar is then obtained as the $c \to 0$ limit of (1) (with the upper sign for the mass term), after having first redefined $\Pi_\Phi$, $\Phi$ and $M$ as

$$\Pi_\Phi = \frac{\pi}{c}, \qquad \Phi = c\,\phi, \qquad M = \frac{m}{c^2}. \tag{2}$$

Explicitly, one finds

$$\mathcal{L}_{\text{el. scalar}} = \pi \partial_t \phi - \frac{1}{2}\pi^2 - \frac{m^2}{2}\phi^2, \quad \text{or equivalently} \quad \mathcal{L}_{\text{el. scalar}} = \frac{1}{2}(\partial_t \phi)^2 - \frac{m^2}{2}\phi^2, \tag{3}$$

where the second Lagrangian is obtained from the first one by eliminating the auxiliary field $\pi$ using its equation of motion. Alternatively, one can first perform the redefinitions

$$\Pi_\Phi = \pi, \qquad \Phi = \phi, \qquad M = \frac{m}{c}, \tag{4}$$

before taking the $c \to 0$ limit in (1). For the tachyonic scalar (i.e., with the lower sign in front of the mass term in (1)), this gives a Lagrangian for a magnetic Carroll scalar:

$$\mathcal{L}_{\text{magn. scalar}} = \pi \partial_t \phi - \frac{1}{2}\partial_a \phi \partial^a \phi + \frac{m^2}{2}\phi^2. \tag{5}$$

Unlike the electric theory, which can be written in terms of a single scalar $\phi$, the magnetic theory involves two scalar fields $\pi$, $\phi$. These transform in a reducible indecomposable representation of the Carroll group, since under Carroll boosts the scalar $\pi$ transforms to $\phi$, but not vice versa [27].

The purpose of this paper is to discuss an analogue of the above electric and magnetic limits for fermion fields. Such models are expected to be relevant, e.g., to describe supergravity theories within flat space holography. Electric and magnetic Carroll fermion field theories have been derived using various limit procedures (e.g., by considering the $c \to 0$ limit of the equations of motion or in a Lagrangian or Hamiltonian formalism) in the literature before [11, 15, 24].[1] Here, we aim to provide a systematic analysis of the possible limits to obtain recently constructed, as well as novel Lagrangians for both electric and magnetic Carroll fermions, i.e., Lagrangians whose field equations can be viewed as 'square roots' of the equations of motion of electric and magnetic Carroll scalars. The systematic nature of our treatment also clarifies some subtleties that arise when considering Carroll limits of fermion field theories.

The procedure presented in section 2 of this paper in particular elucidates how Lagrangians for electric and magnetic Carroll fermions can be obtained from different limits of a single relativistic starting point, not unlike how the electric and magnetic scalar Lagrangians (3) and (5) correspond to different limits of (1). Due to the first-order nature of fermion field Lagrangians, achieving this is slightly more subtle than in the scalar field case. Indeed, since the Klein-Gordon Lagrangian is of second order in derivatives, $\Phi$ and its conjugate momentum $\Pi_\Phi$ in (1) are two independent phase-space variables. They can thus naturally be rescaled independently (as in (2) and (4)), giving rise to two different Carroll limits. By contrast, since a relativistic fermion $\Psi$ obeys a first-order equation of motion, its conjugate momentum $\Pi_\Psi$ is no longer an independent variable, but obeys a primary second-class constraint that says that $\Pi_\Psi$ is proportional to $\Psi^\dagger$. Any rescaling of $\Psi$ then fixes a corresponding rescaling of $\Pi_\Psi$ and consequently only one regular[2] Carroll limit of the Dirac action can be defined that turns out to be an electric one.

In order to be able to take both electric and magnetic limits in a regular manner, we start in subsection 2.2 from an off-diagonal Lagrangian for two relativistic Dirac spinors. We furthermore decompose these two fermion fields in two spinors that each constitute a representation of the spatial rotation subgroup of the Lorentz group. The four spinors thus obtained give enough freedom to perform the independent rescalings needed to take not only an electric Carroll limit, but also a magnetic one that features an indecomposable reducible spinor representation of the Carroll group. While these limits can be taken in arbitrary dimensions, they do not always lead to a minimal result, in the sense that sometimes one can perform a further consistent truncation that still gives a non-trivial theory for an electric or magnetic Carroll fermion. For instance, in even dimensions, a consistent truncation on our result for the magnetic Carroll limit leads to a novel action[3] for a minimal magnetic Carroll fermion, with as many components as a single Dirac spinor. In addition to describing the two Carroll limits in a Lagrangian formalism involving two spinors, we will in subsection 2.3 also outline how the resulting theories can be obtained via a singular limit of the usual Dirac action in Hamiltonian form, along the lines already explored in [24, 32].

Once a detailed description of Carroll limits for spinors in flat space is achieved, the extension to fermions in curved spacetime and coupled to gravity does not present particular difficulties and we illustrate this in two particular cases of electric and magnetic limits in section 3. In particular, we discuss the coupling of the electric and minimal magnetic Carroll fermions

---

[1]An interesting bottom-up construction of Carroll fermion field theories appeared in [18, 19] (see also [28]). In this approach, one does not start from a $c \to 0$ limit of relativistic fermions, but instead constructs spinor representations of the homogeneous Carroll group, starting from degenerate Clifford algebras [29–31].

[2]Rescaling $\Pi_\Psi$ and $\Psi$ in a way that violates the primary constraint that relates them can lead to a second Carroll limit. We discuss such a limit in subsection 2.3, but we consider it as singular, since the resulting theory contains twice as many fermionic degrees of freedom as the original relativistic starting point, due to the fact that in this limit $\Pi_\Psi$ and $\Psi$ are treated independently.

[3]While this action has to our knowledge not yet appeared in the literature, the field equations stemming from it have been considered before in the massless case in [11].

to a first-order formulation of Carroll gravity, called magnetic Carroll gravity, emphasising the role of the spin-connection field. We show that, in passing to a second-order formulation, only magnetic Carroll fermions lead to quartic interactions for fermions in the action. Taking the limit directly in a second-order formulation of General Relativity coupled to fermions, leads to an additional divergent term in the action. Accepting this leading order term as the basic invariant leads to a second-order formulation of Carroll gravity, called electric Carroll gravity, that is independent of the spin-connection and does not contain any surviving fermions. Alternatively, one may control this divergent term by applying a Hubbard-Stratonovich transformation after which the sub-leading terms basically lead to the same coupling of magnetic Carroll gravity to Carroll fermions as found before.

## 2  Carroll Fermions

In this section, we will discuss how electric and magnetic Carroll fermions arise as limits of relativistic fermions, in which the speed of light $c$ is sent to zero. We will first consider the $c \to 0$ limit of the Lorentz transformation rules of a relativistic Dirac spinor in any spacetime dimensions. We will show that this limit can be taken in two different ways, leading to Carroll transformation rules of electric and magnetic Carroll fermions respectively. Next, in subsection 2.2, we will consider two different ways of taking the $c \to 0$ limit on an off-diagonal Lagrangian for two Dirac spinors and show that, after performing suitable consistent truncations, this leads to Lagrangians for electric and magnetic Carroll fermions.[4] Finally, in subsection 2.3 we will reconsider the magnetic Carroll fermion action from a Hamiltonian point of view, as a singular limit of the usual Dirac action in Hamiltonian form.

### 2.1  Carroll transformations of fermion fields

To obtain the transformation rules of fermionic fields under (homogeneous) Carroll symmetries, we start from a relativistic Dirac spinor $\Psi$ in $D$-dimensional Minkowski spacetime with inertial coordinates $X^A$, $A = 0, 1, ..., D-1$. Under infinitesimal Lorentz transformations (with parameters $\Lambda^{AB} = -\Lambda^{BA}$), $\Psi$ transforms as follows:

$$\delta\Psi = \Xi^A \frac{\partial \Psi}{\partial X^A} - \frac{1}{4}\Lambda^{AB}\Gamma_{AB}\Psi, \tag{6}$$

with

$$\delta X^A \equiv X'^A - X^A = -\Xi^A, \qquad \Xi^A = \Lambda^A{}_B X^B, \tag{7}$$

and where $\Gamma_{AB} = \frac{1}{2}[\Gamma_A, \Gamma_B]$ is the usual antisymmetric combination of gamma matrices realising the Lorentz algebra. The Carroll limit formally consists of sending the speed of light $c$ to 0.[5] It is however convenient to first define $\tilde{c} = 1/c$, which has opposite dimension, and instead take $\tilde{c} \to \infty$. The Carroll limit is then defined by first performing the following redefinition of the coordinates and parameters:

$$X^0 = \frac{t}{\tilde{c}}, \qquad X^a = x^a, \qquad \Lambda^{ab} = \lambda^{ab}, \qquad \Lambda^{0a} = \frac{1}{\tilde{c}}\lambda^{0a}, \tag{8}$$

and next taking the limit $\tilde{c} \to \infty$. Note that in our conventions both the Minkowski metric and the gamma matrices are dimensionless, so that we will not rescale them. Applying this

---

[4]In the electric case, the truncation is allowed in any dimension $D$, whereas in the magnetic case, the truncation is allowed only in even $D$, as it requires the use of the $\Gamma_\star$ matrix defined in Appendix A.

[5]Strictly speaking, one should first introduce a dimensionless parameter $\omega$ via the redefinition $c \to \omega c$ and then take the limit $\omega \to 0$.

procedure to (7) leads to the usual transformations of the spacetime coordinates under spatial rotations and Carroll boosts with parameters $\lambda^{ab}$ and $\lambda^{0a}$ respectively,

$$\delta t \equiv t' - t = -\xi^0, \qquad \delta x^a \equiv x'^a - x^a = -\xi^a, \quad \text{with} \quad \xi^0 = \lambda^0{}_a x^a, \quad \text{and} \quad \xi^a = \lambda^a{}_b x^b. \tag{9}$$

The $\tilde{c} \to \infty$ limit, applied to the Lorentz transformation rule (6) after performing the redefinitions (8), gives the following transformation rule of a fermion $\psi \equiv \Psi$:

$$\delta\psi = \xi^0 \frac{\partial \psi}{\partial t} + \xi^a \frac{\partial \psi}{\partial x^a} - \frac{1}{4} \lambda^{ab} \Gamma_{ab} \psi. \tag{10}$$

We note that only spatial rotations appear in the spin part of the transformation rule of the above Carroll fermion. As we will see in the next subsection, this transformation behaviour is characteristic of an electric Carroll fermion. It turns out that there exists a second way of taking the limit that leads to a transformation that contains Carroll boosts in its spin part and that is the relevant one for magnetic Carroll fermions. This limit uses the following orthogonal projection operators[6]

$$P_\pm \equiv \frac{1}{2}\left(\mathbb{1} \pm \mathrm{i}\Gamma^0\right), \quad \text{obeying} \quad P_\pm P_\pm = P_\pm, \quad P_\pm P_\mp = 0, \quad P_\pm^\dagger = P_\pm. \tag{11}$$

Using these projectors, one can split a Dirac spinor $\Psi$ into two independent components $\Psi_\pm \equiv P_\pm \Psi$, each of which transforms covariantly with respect to spatial rotations because $[\Gamma_0, \Gamma_{ab}] = 0$. On the other hand, this projection allows one to distinguish between rotations and boosts since $[\Gamma_0, \Gamma_{0b}] \neq 0$, thus allowing for a non-trivial action of the latter on the fields in the $\tilde{c} \to \infty$ limit. This is achieved by redefining $\Psi_\pm$ differently when defining a Carroll limit. More precisely, we make the following redefinition:

$$\Psi_\pm = \tilde{c}^{\pm 1/2 + \epsilon} P_\pm \psi_\pm, \tag{12}$$

where $\epsilon$ is a free parameter. Note that this redefinition is invertible:

$$\psi_\pm = \tilde{c}^{\mp 1/2 - \epsilon} P_\pm \Psi_\pm. \tag{13}$$

Substituting the redefinitions (8) and (12) into the Lorentz transformation rule (6) we obtain the following transformation rule of a projected Carroll fermion:

$$\delta\psi_+ = \xi^0 \frac{\partial \psi_+}{\partial t} + \xi^a \frac{\partial \psi_+}{\partial x^a} - \frac{1}{4} \lambda^{ab} \Gamma_{ab} \psi_+, \tag{14a}$$

$$\delta\psi_- = \xi^0 \frac{\partial \psi_-}{\partial t} + \xi^a \frac{\partial \psi_-}{\partial x^a} - \frac{1}{4} \lambda^{ab} \Gamma_{ab} \psi_- - \frac{1}{2} \lambda^{0a} \Gamma_{0a} \psi_+. \tag{14b}$$

Note that the Carroll boosts now appear non-trivially in the spin part of this transformation rule. Even if our Clifford algebra is still the usual relativistic one, the projected spinors $\psi_\pm$ form a reducible but indecomposable representation of the homogeneous Carroll group (that consists of spatial rotations and Carroll boosts). This is because only a subset of the Lorentz generators enters the transformation rule of some fermionic components after taking the $\tilde{c} \to \infty$ limit.

---

[6]Note that for even $D$ one can also split a Dirac spinor in another way that preserves spatial rotations by using the projectors $\tilde{P}_\pm = \frac{1}{2}\left(\mathbb{1} \pm \Gamma^0 \Gamma_\star\right)$, with $\Gamma_\star$ defined in (A.3). However, it turns out that using these $\tilde{P}_\pm$ projectors leads to similar results that are mapped to the ones using the $P_\pm$ projectors via the gamma matrix isomorphism $\Gamma^A \to \pm i \Gamma^A \Gamma_\star$.

## 2.2 Electric and magnetic Carroll fermions from a Lagrangian perspective

Having defined the two limits leading to a Carroll fermion and a projected Carroll fermion, we now take analogous limits at the level of the Lagrangian. We will use as a starting point an off-diagonal Lagrangian for two Dirac spinors $\Psi$ and $\mathbf{X}$. This choice is motivated by the goal of describing in a single framework both limits, although, as we will discuss later, an action describing an electric Carroll fermion can be defined simply by taking the $\tilde{c} \to \infty$ limit of the usual Dirac's action. The off-diagonal Lagrangian reads

$$\mathcal{L}_{\text{off-diag}} = \bar{\mathbf{X}} \Gamma^A \partial_A \Psi - \frac{M}{\tilde{c}} \bar{\mathbf{X}} \Psi + \text{h.c.}, \tag{15}$$

where $M$ is a complex parameter with the dimension of mass and where the Dirac conjugate is defined by $\bar{\Psi} \equiv i \Psi^\dagger \Gamma^0$.[7] Both Dirac spinors $\Psi$ and $\mathbf{X}$ transform under Lorentz transformations as in equations (6) and (7). As it is, the Lagrangian (15) describes both tachyonic and non-tachyonic modes, due to the fact the parameter $M$ can be chosen to be complex.[8] While this Lagrangian may seem unconventional, we note that in the next subsection we will argue that our results on its Carrollian limits can also be reproduced from a more conventional starting point, namely using a Dirac action in Hamiltonian form.

It is clear that when we define a limit in which we naïvely scale all components of the Dirac spinors as $\mathbf{X} \to \tilde{c}^\alpha \mathbf{X}$ and $\Psi \to \tilde{c}^\beta \Psi$, the terms in the action with a time derivative will always dominate the terms with a spatial derivative due to the additional factor of $\tilde{c}$ brought by $\partial_0 = \tilde{c} \, \partial_t$. In analogy to the bosonic case, we will call such a limit an *electric Carroll* limit. To create more freedom in performing rescalings of the spinors we introduce the following projected spinors:

$$\Psi_\pm = P_\pm \Psi, \qquad \mathbf{X}_\pm = P_\pm \mathbf{X}, \tag{16}$$

where $P_\pm$ are the projectors that we introduced in (11). In terms of the projected spinors the Lagrangian (15) reads

$$\mathcal{L}_{\text{off-diag}} = \tilde{c} \left( \bar{\mathbf{X}}_+ \Gamma^0 \dot{\Psi}_+ + \bar{\mathbf{X}}_- \Gamma^0 \dot{\Psi}_- \right) + \bar{\mathbf{X}}_+ \Gamma^a \partial_a \Psi_- + \bar{\mathbf{X}}_- \Gamma^a \partial_a \Psi_+ - \frac{M}{\tilde{c}} \left( \bar{\mathbf{X}}_+ \Psi_+ + \bar{\mathbf{X}}_- \Psi_- \right) + \text{h.c.}, \tag{17}$$

where we used the notation $\dot{\Psi} \equiv \partial \Psi / \partial t$. The four spinors $\Psi_\pm$ and $\mathbf{X}_\pm$, which we are free to scale independently, give us sufficient freedom to define a second limit that leads to an action resembling the magnetic scalar case. We will call this limit a *magnetic Carroll* limit. Below we discuss these two different limits in more detail.

**The electric Carroll limit**

As anticipated, an electric Carroll limit can be defined by combining the rescalings (8) with any rescaling of the spinors of the form $\mathbf{X} \to \tilde{c}^\alpha \mathbf{X}$ and $\Psi \to \tilde{c}^\beta \Psi$. Alternatively, one can even rescale the four projected spinors as in (12),

$$\Psi_+ = \sqrt{\tilde{c}} \, \tilde{c}^\epsilon \psi_+, \qquad \Psi_- = \frac{1}{\sqrt{\tilde{c}}} \, \tilde{c}^\epsilon \psi_-, \tag{18a}$$

$$\mathbf{X}_+ = \sqrt{\tilde{c}} \, \tilde{c}^\epsilon \chi_+, \qquad \mathbf{X}_- = \frac{1}{\sqrt{\tilde{c}}} \, \tilde{c}^\epsilon \chi_-, \tag{18b}$$

---

[7] More details about our notation and conventions can be found in Appendix A.

[8] This is similar to the bosonic case, where the starting point of certain Carroll limits also requires a tachyonic mode. An additional feature in the fermionic case is that, after diagonalisation, one of the two spinors has a negative definite kinetic term.

and take $\epsilon = -1$ in order to have no overall rescaling of the Lagrangian. Furthermore, we rescale the complex parameter $M$ as follows:[9]

$$M = \tilde{c}^{\alpha} m\,, \tag{19}$$

and we take $\alpha = 2$. By taking this electric Carroll limit, the spinors $\chi_-$ and $\psi_-$ drop out[10] and we obtain the following off-diagonal Lagrangian in terms of $\chi_+$ and $\psi_+$:

$$\mathcal{L}_{\text{off-diag}} = \bar{\chi}_+ \Gamma^0 \dot{\psi}_+ - m \bar{\chi}_+ \psi_+ + \text{h.c.} \tag{20}$$

The latter is invariant under the following Carroll transformations:[11]

$$\delta \psi_+ = \xi^0 \dot{\psi}_+ + \xi^a \partial_a \psi_+ - \frac{1}{4} \lambda_{ab} \Gamma^{ab} \psi_+\,, \tag{21a}$$

$$\delta \chi_+ = \xi^0 \dot{\chi}_+ + \xi^a \partial_a \chi_+ - \frac{1}{4} \lambda_{ab} \Gamma^{ab} \chi_+\,, \tag{21b}$$

that only involve the generators of spatial rotations as in (10) as expected in an electric limit. Furthermore, it involves the same total number of spinor components as Dirac's Lagrangian. On the other hand, upon diagonalisation it displays kinetic terms with opposite signs as in the Lagrangian (15). The transformation rules (21) show however that it is consistent to perform the following truncation

$$\chi_+ = \psi_+\,, \tag{22}$$

after which we obtain the following electric Carroll Lagrangian in diagonal form [15, 19]

$$\mathcal{L}_{\text{electric Carroll}} = 2\bar{\psi}_+ \Gamma^0 \dot{\psi}_+ - 2\,\mathfrak{Re}(m)\bar{\psi}_+ \psi_+\,, \tag{23}$$

where $\mathfrak{Re}(m) = \frac{1}{2}(m + m^*)$ is a real mass parameter.

As we mentioned at the beginning of this section, we introduced the parent Lagrangian (15) to describe both electric and magnetic limits within the same framework.[12] However, if one wants to just obtain an electric Carroll Lagrangian or, more specifically the minimal electric Carroll Lagrangian (23), one can just start from the Dirac Lagrangian, corresponding to the truncation $\mathbf{X} = \Psi$ in the parent Lagrangian (15). One next redefines $\Psi = \tilde{c}^{-1/2}\psi$ and takes the Carroll limit. In this setup it is not necessary to introduce a truncation only to the $\psi_+$ sector, but after taking this limit, one has the right to impose a further truncation by setting $\psi_- = 0$, thus obtaining the minimal electric Carroll action (23) (where the truncation was needed to avoid ghost propagation). In section 3, we will show how a curved space analogue of the plain way of taking the electric Carroll limit leads to an action for an electric Carroll fermion coupled to Carroll gravity.

---

[9]Note that we work in the convention where $M$ has the dimension of a mass parameter, and therefore different values of $\alpha$ lead to parameters $m$ with different dimensions. In order to avoid spurious notation, we will keep calling this parameter $m$.

[10]The choice of of keeping $\psi_+$ and $\chi_+$ is purely conventional: we could have kept $\psi_-$ and $\chi_-$ by rescaling the fields in the opposing way or even keep all components by rescaling all spinors with the same power of $\tilde{c}$. The splitting induced by $P_\pm$ is indeed not necessary to define the electric limit, while it will be crucial to define the magnetic limit.

[11]Since it describes a free field theory, we expect that the Lagrangian (20) has additional symmetries, like it happens in the bosonic case for Carrollian scalars, both in two dimensions [33, 34] and generic $D$ [11, 20, 35, 36].

[12]For bosonic theories, it is also possible to obtain their respective electric (magnetic) Carroll actions in a unified way as the leading (subleading) terms of the $c$-expansion of the relativistic action [13]. To show this, one is however required to use a Hubbard-Stratonovich transformation, which can only be performed if the divergent term is a square. Due to the first-order nature of the Dirac Lagrangian, the divergent fermionic kinetic term cannot be written as a square. This is the main difference with the bosonic case, hindering a straightforward application of the $c$-expansion method to fermionic theories.

**The magnetic Carroll limit**

The magnetic Carroll limit is defined by the rescalings (8) together with the following 'twisted' rescalings of the four spinors $\Psi_\pm$ and $\mathbf{X}_\pm$:

$$\Psi_+ = \sqrt{\tilde{c}}\,\tilde{c}^\epsilon\psi_+\,, \qquad\qquad \Psi_- = \frac{1}{\sqrt{\tilde{c}}}\,\tilde{c}^\epsilon\psi_-\,, \tag{24a}$$

$$\mathbf{X}_+ = \frac{1}{\sqrt{\tilde{c}}}\,\tilde{c}^\epsilon\chi_+\,, \qquad\qquad \mathbf{X}_- = \sqrt{\tilde{c}}\,\tilde{c}^\epsilon\chi_-\,. \tag{24b}$$

We now take $\epsilon = -1/2$, in order to have no overall rescaling of the Lagrangian. Furthermore, we rescale the complex parameter $M$ as in (19) with $\alpha = 2$. By taking this magnetic limit, all four spinor projected components survive, and we obtain the following action:

$$\mathcal{L}_{\text{off-diag}} = \bar{\chi}_+\Gamma^0\dot{\psi}_+ + \bar{\chi}_-\Gamma^0\dot{\psi}_- + \bar{\chi}_-\Gamma^a\partial_a\psi_+ - m(\bar{\chi}_+\psi_+ + \bar{\chi}_-\psi_-) + \text{h.c.} \tag{25}$$

This action is invariant under the following Carroll transformations:

$$\delta\psi_+ = \xi^0\dot{\psi}_+ + \xi^a\partial_a\psi_+ - \frac{1}{4}\lambda_{ab}\Gamma^{ab}\psi_+\,, \tag{26a}$$

$$\delta\psi_- = \xi^0\dot{\psi}_- + \xi^a\partial_a\psi_- - \frac{1}{4}\lambda_{ab}\Gamma^{ab}\psi_- - \frac{1}{2}\lambda_{0a}\Gamma^{0a}\psi_+\,, \tag{26b}$$

$$\delta\chi_+ = \xi^0\dot{\chi}_+ + \xi^a\partial_a\chi_+ - \frac{1}{4}\lambda_{ab}\Gamma^{ab}\chi_+ - \frac{1}{2}\lambda_{0a}\Gamma^{0a}\chi_-\,, \tag{26c}$$

$$\delta\chi_- = \xi^0\dot{\chi}_- + \xi^a\partial_a\chi_- - \frac{1}{4}\lambda_{ab}\Gamma^{ab}\chi_-\,. \tag{26d}$$

These transformation rules show that in even dimensions we can make the following truncations:[13]

$$\chi_\pm = \Gamma_\star\psi_\mp\,, \tag{27}$$

where we have made use of the $\Gamma_\star$ matrix[14]

$$\Gamma_\star = (-i)^{\frac{D}{2}+1}\Gamma^0\Gamma^1\cdots\Gamma^{D-1}\,, \tag{28}$$

to connect the different projections. Upon making this truncation, we obtain the following minimal Lagrangian:[15]

$$\mathcal{L}_{\text{magnetic Carroll, 1}} = 2\bar{\psi}_-\Gamma^0\Gamma_\star\dot{\psi}_+ + 2\bar{\psi}_+\Gamma^0\Gamma_\star\dot{\psi}_- + 2\bar{\psi}_+\Gamma^a\Gamma_\star\partial_a\psi_+$$
$$+ 2i\,\mathfrak{Im}(m)(\bar{\psi}_+\Gamma_\star\psi_- + \bar{\psi}_-\Gamma_\star\psi_+)\,, \tag{29}$$

where $\mathfrak{Im}(m) = -\frac{i}{2}(m - m^*)$ is a real mass parameter. In the massless case, the equations of motion stemming from this Lagrangian correspond to those obtained in [11] by taking a

---

[13]Note that we give here two truncations. One could also impose only one of the two truncations. We will not consider this non-minimal case further here. Note also that the parent action (15) is invariant under the usual action of parity on spinor fields. Our truncation is thus breaking parity, although it might be possible to define a different action of parity after the Carrollian limit.

[14]Further properties of this $\Gamma_\star$ matrix can be found in Appendix A.

[15]There is also a second way of obtaining the magnetic Carroll 1 Lagrangian (29). This consists in replacing the off-diagonal mass term in the parent Lagrangian (15) with the following diagonal one

$$\mathcal{L}_{\text{mass-diag}} = -i\frac{M}{2\tilde{c}}\left(\bar{\Psi}\Gamma_\star\Psi - \bar{\mathbf{X}}\Gamma_\star\mathbf{X}\right)\,,$$

where here $M$ is a real parameter. Then, by applying the same magnetic rescaling defined above, and by rescaling $M = \tilde{c}^2 m$, one gets (29).

limit of the Dirac equation but, to our knowledge, an action principle involving this minimal projected Carroll fermion has not appeared before.

One can obtain a second magnetic Carroll Lagrangian with a different mass term by inserting a $\Gamma_\star$ in the mass term of the initial Lagrangian (15).[16] This leads to the following alternative mass term:

$$\mathcal{L}_{\text{mass}} = -\frac{M}{\tilde{c}} \bar{X} \Gamma_\star \Psi + \text{h.c.} \tag{30}$$

After taking the magnetic limit defined above, and rescaling the mass parameter $M$ as in (19) with $\alpha = 1$, this mass term becomes

$$\mathcal{L}_{\text{mass}} = -m \, \bar{\chi}_- \Gamma_\star \psi_+ + \text{h.c.}, \tag{31}$$

which after performing the truncation (27) reduces to the mass term of a second magnetic Carroll Lagrangian:[17]

$$\mathcal{L}_{\text{magnetic Carroll, 2}} = 2\bar{\psi}_- \Gamma^0 \Gamma_\star \dot{\psi}_+ + 2\bar{\psi}_+ \Gamma^0 \Gamma_\star \dot{\psi}_- + 2\bar{\psi}_+ \Gamma^a \Gamma_\star \partial_a \psi_+ - 2\,\mathfrak{Re}(m)\bar{\psi}_+ \psi_+. \tag{32}$$

It is also possible to obtain the two magnetic Carroll actions (29) and (32) by taking the limit directly on a Lagrangian depending only on a single Dirac spinor $\Psi$. A way to do it, is to either truncate the parent Lagrangian (15) or the same Lagrangian but with modified mass term (30) by imposing $X_\pm = \Gamma_\star \Psi_\mp$. This leads to the following 'tachyonic Dirac'[18] Lagrangians

$$\mathcal{L} = \bar{\Psi} \Gamma^A \Gamma_\star \partial_A \Psi - i\frac{M}{\tilde{c}} \bar{\Psi} \Gamma_\star \Psi, \quad \text{and} \quad \mathcal{L} = \bar{\Psi} \Gamma^A \Gamma_\star \partial_A \Psi - \frac{M}{\tilde{c}} \bar{\Psi} \Psi, \tag{33}$$

where the mass parameter $M$ is now assumed to be real. Then one needs to rescale $\Psi_\pm$ as in (24a) and take the Carroll limit to recover (29) and (32). In section 3, we will obtain the action of a magnetic Carroll fermion, coupled to Carroll gravity, as a curved space analogue of this way of taking the magnetic Carroll limit.

This concludes our discussion of Carroll fermions and Carroll limits. We have summarised some of our findings in Table 1.

## 2.3 Carroll limit of Dirac's action in Hamiltonian form

We now show that the previous magnetic Carrollian Lagrangian (29) can also be recovered by considering a limit of the Dirac Lagrangian

$$\mathcal{L} = \frac{1}{2} \bar{\Psi} \Gamma^A \partial_A \Psi - \frac{1}{2} \partial_A \bar{\Psi} \Gamma^A \Psi - \frac{\mathfrak{Re}(M)}{\tilde{c}} \bar{\Psi} \Psi, \tag{34}$$

rewritten in Hamiltonian form, where the notation $\mathfrak{Re}(M)$ anticipates that later we will introduce an imaginary mass parameter. To proceed, one can introduce the conjugate momenta

$$\Pi \equiv \frac{\partial \mathcal{L}}{\partial \dot{\Psi}} = \frac{\tilde{c}}{2} \bar{\Psi} \Gamma^0, \qquad \bar{\Pi} \equiv \frac{\partial \mathcal{L}}{\partial \dot{\bar{\Psi}}} = -\frac{\tilde{c}}{2} \Gamma^0 \Psi, \tag{35}$$

---

[16]A similar trick does not work in the electric case since in there we only have a kinetic term involving two spinors of the same chirality. By inserting a $\Gamma_\star$ in the mass term, one introduces a spinor with opposite chirality that acts as a Lagrange multiplier which on-shell sets to zero the action. Also, as explained in footnote 6, inserting a $\Gamma_\star$ into the projection operators leads to Lagrangians that are equivalent to the Lagrangians we obtained using projection operators without a $\Gamma_\star$.

[17]As we mentioned for the magnetic Carroll 1 Lagrangian, there is also a second way of obtaining the magnetic Carroll 2 Lagrangian (32). The diagonal mass term to consider in this case is

$$\mathcal{L}_{\text{mass-diag}} = -\frac{M}{2\tilde{c}} \left( \bar{\Psi} \Psi - \bar{X} X \right),$$

and the Carroll limit requires to rescale $M = \tilde{c} \, m$.

[18]That these two Lagrangians describe tachyonic fermions is seen by multiplying their equations of motion with the operators $\Gamma^A \Gamma_\star \partial_A - i(M/\tilde{c})\Gamma_\star$ and $\Gamma^A \Gamma_\star \partial_A + (M/\tilde{c})$ respectively. This leads to the Klein-Gordon equation $(\Box + M^2/\tilde{c}^2)\Psi = 0$ for a tachyon.

Table 1: Summary of the minimal electric and magnetic Carroll Lagrangians. The number $\alpha$ is defined as in equation (19), while the number $\epsilon$ is defined in (12). The last column indicates the equation number where the expressions for the different Lagrangians can be found.

| Model | $\alpha$ | $\epsilon$ | equation |
|---|---|---|---|
| electric Carroll | 2 | $-1$ | (23) |
| magnetic Carroll 1 | 2 | $-1/2$ | (29) |
| magnetic Carroll 2 | 1 | $-1/2$ | (32) |

that satisfy the conjugation relations $\bar{\Pi} = i\Gamma^0\Pi^\dagger, \Pi = i\bar{\Pi}^\dagger\Gamma^0$. The Dirac action can then be rewritten as

$$\mathcal{L}[\Psi, \Pi, \lambda] = \Pi\dot{\Psi} + \dot{\bar{\Psi}}\bar{\Pi} - \mathcal{H} + \left(\Pi - \frac{\tilde{c}}{2}\bar{\Psi}\Gamma^0\right)\lambda + \bar{\lambda}\left(\bar{\Pi} + \frac{\tilde{c}}{2}\Gamma^0\Psi\right), \tag{36}$$

where we have introduced Lagrange multipliers $\lambda$ and $\bar{\lambda}$ that are related via Dirac conjugation ($\bar{\lambda} = i\lambda^\dagger\Gamma^0$), and where the Hamiltonian density is given by

$$\mathcal{H} = -\frac{1}{2}\bar{\Psi}\Gamma^a\partial_a\Psi + \frac{1}{2}\partial_a\bar{\Psi}\Gamma^a\Psi + \frac{\mathfrak{Re}(M)}{\tilde{c}}\bar{\Psi}\Psi. \tag{37}$$

The equations of motion of the Lagrange multipliers correspond to the following second class constraints for the canonical variables

$$\zeta_1 \equiv \Pi - \frac{\tilde{c}}{2}\bar{\Psi}\Gamma^0 \approx 0, \qquad \zeta_2 \equiv \bar{\Pi} + \frac{\tilde{c}}{2}\Gamma^0\Psi \approx 0. \tag{38}$$

Equivalently, performing the following redefinition of the Lagrange multiplier $\lambda$

$$\lambda = \lambda' + \left(-\frac{1}{\tilde{c}}\Gamma^0\Gamma^a\partial_a + \frac{1}{\tilde{c}^2}\mathfrak{Re}(M)\Gamma^0\right)\Psi + \frac{i}{\tilde{c}^2}\mathfrak{Im}(M)\Gamma^0\Psi, \tag{39}$$

in the Lagrangian (36), one can rewrite the latter as

$$\mathcal{L}[\Psi, \Pi, \lambda'] = \Pi\dot{\Psi} + \dot{\bar{\Psi}}\bar{\Pi} - \mathcal{H}' + \left(\Pi - \frac{\tilde{c}}{2}\bar{\Psi}\Gamma^0\right)\lambda' + \bar{\lambda}'\left(\bar{\Pi} + \frac{\tilde{c}}{2}\Gamma^0\Psi\right), \tag{40}$$

with the Hamiltonian

$$\mathcal{H}' = \frac{1}{\tilde{c}}\Pi\Gamma^0\Gamma^a\partial_a\Psi + \frac{1}{\tilde{c}}\partial_a\bar{\Psi}\Gamma^a\Gamma^0\bar{\Pi} - \frac{M}{\tilde{c}^2}\Pi\Gamma^0\Psi + \frac{M^*}{\tilde{c}^2}\bar{\Psi}\Gamma^0\bar{\Pi}. \tag{41}$$

Note that the field redefinition (39) contains a new real parameter, $\mathfrak{Im}(M)$, that has been introduced so as to obtain a mass term with the same structure as that in (15), depending on a complex mass parameter $M$. The Hamiltonian (37) and the one-parameter family of Hamiltonians (41), parameterised by $\mathfrak{Im}(M)$, are clearly equivalent for any finite value of $\tilde{c}$. The parameter $\mathfrak{Im}(M)$ does not play any role at finite $\tilde{c}$ and it can always be eliminated by undoing the field redefinition (39). On the other hand, different rewritings of the Hamiltonian will lead to different Carroll actions because we will define the magnetic limit so that $\lambda$ disappears when sending $\tilde{c} \to \infty$.

Since one is allowed to eliminate $\Pi$ and $\bar{\Pi}$ using the algebraic equations of motion (38), one readily sees that the Lagrangian (40) (or (36)) is equivalent to the usual Dirac Lagrangian

(34). We can then equally well consider the magnetic Carroll limit of the Lagrangian (40) with $\mathcal{H}'$ given by (41). As in the previous section, we introduce the projected spinors

$$\Psi_\pm = P_\pm \Psi, \qquad \Pi_\pm = \Pi P_\pm, \qquad \lambda'_\pm = P_\pm \lambda', \qquad P_\pm = \frac{1}{2}(\mathbb{1} \pm i\Gamma^0), \tag{42}$$

and we take the $\tilde{c} \to \infty$ limit of (40) after imposing the following rescalings of the canonical variables

$$\Pi_+ = \frac{1}{\tilde{c}}\pi_+, \qquad \Psi_+ = \tilde{c}\,\psi_+, \tag{43a}$$

$$\Pi_- = \pi_-, \qquad \Psi_- = \psi_-, \tag{43b}$$

as well as the rescalings

$$\lambda'_+ = \frac{1}{\tilde{c}^{3+\epsilon}}\tilde{\lambda}_+, \quad \lambda'_- = \frac{1}{\tilde{c}^{2+\epsilon}}\tilde{\lambda}_-, \quad \text{with} \quad \epsilon \geq 0. \tag{44}$$

We further rescale the complex mass parameter as $M = \tilde{c}^2 m$. The rescalings (43) have been chosen so as to keep the kinetic term $\Pi\dot{\Psi} + \dot{\bar{\Psi}}\bar{\Pi}$ untouched in the limit, in agreement with the strategy adopted in the bosonic case in [12]. Due to the disappearance of the Lagrange multipliers in the limit, the second class constraints $\zeta_1$ and $\zeta_2$ are lost and the fields $\Pi$ and $\Psi$ become independent. In the Carroll limit, we thus effectively doubled the number of fields with respect to the original Dirac action and in the massive case we introduced a new parameter, the imaginary part of $M$.

In the limit $\tilde{c} \to \infty$, the Lagrangian (40) with $\mathcal{H}'$ given in (41) takes the form

$$\begin{aligned}
\mathcal{L} = {} &\pi_+\dot{\psi}_+ + \pi_-\dot{\psi}_- + \dot{\bar{\psi}}_+\bar{\pi}_+ + \dot{\bar{\psi}}_-\bar{\pi}_- - \pi_-\Gamma^0\Gamma^a\partial_a\psi_+ - \partial_a\bar{\psi}_+\Gamma^a\Gamma^0\bar{\pi}_- \\
&+ m\pi_+\Gamma^0\psi_+ + m\pi_-\Gamma^0\psi_- - m^*\bar{\psi}_+\Gamma^0\bar{\pi}_+ - m^*\bar{\psi}_-\Gamma^0\bar{\pi}_-,
\end{aligned} \tag{45}$$

which corresponds to the Lagrangian (25), that we obtained starting from the two-fermion relativistic Lagrangian (15), upon performing the identification

$$\pi_\pm = \bar{\chi}_\pm\Gamma^0. \tag{46}$$

To show that (45) is invariant under Carroll symmetries, one needs to show that the Poisson brackets of its energy density vanish, as suggested in [12]. However, we do not need to check this, since (45) matches (25), which is Carroll invariant.[19] In even spacetime dimensions, the same truncation as in (27) can then be introduced to describe a minimal magnetic theory. The approaches presented in this subsection and in the previous one are therefore completely equivalent if one wishes to produce an action principle for a magnetic Carroll theory.

To conclude, we wish to point out that one can also start from the Lagrangian (36) with the Hamiltonian written in the form (37) and rescale the fields as

$$\Pi = \pi, \qquad \Psi = \psi, \qquad \lambda = \frac{1}{\tilde{c}^{2+\epsilon}}\tilde{\lambda}, \quad \text{with} \quad \epsilon \geq 0, \tag{47}$$

therefore avoiding the splitting of the fields into irreducible $SO(D-1)$ representations as in (43). In the limit $\tilde{c} \to \infty$, with the mass rescaled as $M = \tilde{c}m$, one obtains the following Carroll Lagrangian for the two spinors $\psi$ and $\bar{\pi}$ [24]:

$$\mathcal{L} = \pi\dot{\psi} + \dot{\bar{\psi}}\bar{\pi} + \bar{\psi}\Gamma^a\partial_a\psi - \mathfrak{Re}(m)\bar{\psi}\psi. \tag{48}$$

---

[19]A non-trivial cancellation occurs when checking that (25) is invariant under the boost transformation. We expect an analogous non-trivial cancellation to be crucial when checking the vanishing of the Poisson brackets of the energy density.

Its equations of motion have a structure similar to that of a magnetic Carroll scalar:

$$\dot{\pi} = \Gamma^a \partial_a \psi - \mathfrak{Re}(m)\psi, \qquad \dot{\psi} = 0. \tag{49}$$

In spite of the homogeneous rescaling of both $SO(D-1)$ irreducible components of each spinor, the presence of two spinors thus still allows for a magnetic behaviour. The option to perform a truncation leading to a system with the same number of components as a single relativistic Dirac fermion is however less evident in this form. Still, it can be recovered also in this setup by introducing a suitable splitting of the spinors in $SO(D-1)$ irreducible components after the limit. The kinetic term of (48) corresponds indeed to that in our non-minimal magnetic Carrollian Lagrangian (25) upon performing the field redefinition

$$\psi = \psi_+ + \chi_-, \qquad \pi = (\bar{\chi}_+ + \bar{\psi}_-)\Gamma^0, \tag{50}$$

which is legitimate since after the limit the spinors $\psi$ and $\pi$ are independent. On the other hand, we note that the mass term has a different form with respect to that in (25) and, in particular, if one implements the truncation (27) it vanishes.

## 3 Coupling to Carroll gravity

In this section, we will discuss the coupling of Carroll fermions to Carrollian gravity (see [16, 17] for discussions on the coupling of Carrollian scalars to arbitrary but curved Carrollian backgrounds and [37] for their coupling to Carrollian gravity). This requires a Cartan formulation of Carroll geometry that can be viewed as a special instance of the so-called $p$-brane non-Lorentzian geometries, studied recently in [38].[20] We will review this description of Carroll geometry in section 3.1. Coupling fermions to General Relativity can be done in two ways: either in a first-order Palatini formulation or directly in the second-order formulation. The difference between these two is that in the Palatini formulation, passing to the second-order formalism leads to quartic fermion terms, stemming from fermion bilinear torsion contributions, that are not present when the coupling is performed directly in the second-order formulation. In this section, we will discuss Carrollian analogues of these two inequivalent ways of coupling fermions to gravity. In section 3.2, we will apply a Carroll limit to General Relativity, coupled to a Dirac and tachyonic Dirac fermion in the Palatini formulation, to obtain an analogous first-order coupling of an electric and magnetic Carroll fermion to the so-called magnetic Carroll gravity [12, 37, 41, 42]. We will also comment on the passage to the second-order formulation explored, e.g., in [13, 23, 43–45]. In section 3.3, we will discuss the Carroll limit of General Relativity, coupled to a (tachyonic) Dirac spinor, directly in the second-order formalism.

### 3.1 Carroll geometry

In order to couple Carroll fermions to gravity, a suitable notion of spin-connection fields for Carroll geometry is needed. A convenient way to introduce these spin-connections is by considering a Carroll limit of the Cartan formulation of Lorentzian geometry. Recall that in the latter, the metric is encoded in the solder form $E_\mu{}^A$ (whose inverse $E_A{}^\mu$ is the Vielbein) that transforms under local Lorentz transformations with parameters $\Lambda^{AB} = -\Lambda^{BA}$ as

$$\delta E_\mu{}^A = -\Lambda^A{}_B E_\mu{}^B. \tag{51}$$

---

[20]More precisely, there exists a duality [39, 40] that relates Carroll geometry to $(D-2)$-brane Galilean geometry that was discussed in detail in [38].

A metric-compatible connection is described by a spin-connection field $\Omega_\mu{}^{AB} = -\Omega_\mu{}^{BA}$, whose Lorentz transformation rule is given by

$$\delta\Omega_\mu{}^{AB} = \partial_\mu\Lambda^{AB} - 2\,\Lambda^{[A}{}_C\Omega_\mu{}^{|C|B]}\,, \tag{52}$$

and that obeys the first Cartan structure equations:

$$T_{\mu\nu}{}^A = 2\,\partial_{[\mu}E_{\nu]}{}^A + 2\,\Omega_{[\mu}{}^{AB}E_{\nu]B}\,, \quad \text{where} \quad T_{\mu\nu}{}^A \text{ is the torsion tensor}\,. \tag{53}$$

Note that these constitute as many equations as there are spin-connection components and that each of these equations contains a spin-connection component. As a consequence, the first Cartan structure equations (53) can be solved for all components of $\Omega_\mu{}^{AB}$ in terms of the (inverse) Vielbein and the torsion tensor components in a unique fashion:

$$\Omega_\mu{}^{AB} = E^{[A|\nu|}\left(2\,\partial_{[\mu}E_{\nu]}{}^{B]} - T_{\mu\nu}{}^{B]}\right) - \frac{1}{2}E_{\mu C}\,E^{A\nu}E^{B\rho}\left(2\,\partial_{[\nu}E_{\rho]}{}^C - T_{\nu\rho}{}^C\right)\,. \tag{54}$$

In order to obtain a Cartan formulation of Carroll geometry, we perform the following rescalings

$$E_\mu{}^0 = \frac{1}{\tilde{c}}\tau_\mu\,, \qquad E_\mu{}^a = e_\mu{}^a\,, \qquad E_0{}^\mu = \tilde{c}\tau^\mu\,, \qquad E_a{}^\mu = e_a{}^\mu\,,$$

$$\Omega_\mu{}^{ab} = \omega_\mu{}^{ab}\,, \qquad \Omega_\mu{}^{a0} = \frac{1}{\tilde{c}}\omega_\mu{}^{a0}\,, \qquad T_{\mu\nu}{}^0 = \frac{1}{\tilde{c}}t_{\mu\nu}{}^0\,, \qquad T_{\mu\nu}{}^a = t_{\mu\nu}{}^a\,,$$

$$\Lambda^{ab} = \lambda^{ab}\,, \qquad \Lambda^{0a} = \frac{1}{\tilde{c}}\lambda^{0a}\,, \tag{55}$$

in the transformation rules (51), (52) and first Cartan structure equations (53) and take the $\tilde{c} \to \infty$ limit. This leads to a Carrollian solder form $(\tau_\mu, e_\mu{}^a)$ and Vielbein $(\tau^\mu, e_a{}^\mu)$ that transform under local spatial rotations (with parameters $\lambda^{ab} = -\lambda^{ba}$) and Carroll boosts (with parameters $\lambda^{0a}$) as

$$\delta\tau_\mu = -\lambda^0{}_a e_\mu{}^a\,, \qquad \delta e_\mu{}^a = -\lambda^a{}_b e_\mu{}^b\,, \qquad \delta\tau^\mu = 0\,, \qquad \delta e_a{}^\mu = \lambda^0{}_a \tau^\mu - \lambda_a{}^b e_b{}^\mu\,. \tag{56}$$

Here and in the following, we have freely raised and lowered the $a$, $b$ indices with a Euclidean Kronecker-delta metric. We will frequently use the Carroll Vielbein $(\tau^\mu, e_a{}^\mu)$ to turn curved $\mu$, $\nu$ indices of form fields into flat 0, $a$ indices; e.g., for a one-form $X_\mu$ and a two-form $X_{\mu\nu}$ we define:

$$X_0 \equiv \tau^\mu X_\mu\,, \qquad X_a \equiv e_a{}^\mu X_\mu\,, \qquad X_{0a} \equiv \tau^\mu e_a{}^\nu X_{\mu\nu}\,, \qquad X_{ab} \equiv e_a{}^\mu e_b{}^\nu X_{\mu\nu}\,. \tag{57}$$

The rescalings (55) and $\tilde{c} \to \infty$ limit furthermore give rise to Carrollian spin-connection fields $\omega_\mu{}^{ab}$ and $\omega_\mu{}^{0a}$ for spatial rotations and Carroll boosts that transform as

$$\delta\omega_\mu{}^{ab} = \partial_\mu\lambda^{ab} - 2\lambda^{[a}{}_c\omega_\mu{}^{|c|b]}\,, \qquad \delta\omega_\mu{}^{0a} = \partial_\mu\lambda^{0a} - \lambda^a{}_b\omega_\mu{}^{0b} - \lambda^0{}_b\omega_\mu{}^{ba}\,. \tag{58}$$

The $\tilde{c} \to \infty$ limit of the Lorentzian first Cartan structure equations (53) corresponds to the following Carrollian analogue:

$$\tau_{\mu\nu} + 2\omega_{[\mu}{}^{0a}e_{\nu]a} = t_{\mu\nu}{}^0\,, \quad \text{with} \quad \tau_{\mu\nu} \equiv 2\partial_{[\mu}\tau_{\nu]}\,, \tag{59a}$$

$$e_{\mu\nu}{}^a + 2\omega_{[\mu}{}^{ab}e_{\nu]b} = t_{\mu\nu}{}^a\,, \quad \text{with} \quad e_{\mu\nu}{}^a \equiv 2\partial_{[\mu}e_{\nu]}{}^a\,, \tag{59b}$$

where $t_{\mu\nu}{}^0 \equiv t_{\mu\nu}{}^\rho\tau_\rho$ and $t_{\mu\nu}{}^a \equiv t_{\mu\nu}{}^\rho e_\rho{}^a$. Contrary to what happens in Lorentzian geometry, it is no longer true that each of these first Cartan structure equations contains a spin-connection

component. In particular, multiplying both sides of (59b) with $\tau^\mu e_b{}^\nu$ and symmetrising in $(ab)$, one finds

$$e_{0(a,b)} = t_{0(a,b)}, \tag{60}$$

where $e_{0b,a} \equiv \tau^\mu e_b{}^\nu e_{\mu\nu a}$ and $t_{0b,a} \equiv \tau^\mu e_b{}^\nu t_{\mu\nu a}$. One thus sees that setting components of $t_{0(a,b)}$ equal to zero no longer solely amounts to specifying a particular connection, but additionally leads to differential constraints on $e_\mu{}^a$ and thus constraints on the geometry of the underlying manifold. Such geometric constraints are called intrinsic torsion constraints and the components of $t_{0(a,b)}$ are referred to as intrinsic torsion components. Intrinsic torsion constraints that are covariant with respect to local spatial rotations and Carroll boosts can be imposed in four different ways, leading to four distinct types of Carroll geometry, called Carroll 1 – Carroll 4:

**Carroll 1:** all intrinsic torsion tensors are non-zero.

**Carroll 2:** $t_{0a}{}^a = 0$.

**Carroll 3:** $t_{0\{a,b\}} = 0$.

**Carroll 4:** $t_{0a}{}^a = t_{0\{a,b\}} = 0$.

Here, $t_{0\{a,b\}}$ denotes the symmetric traceless part (in $(ab)$) of $t_{0a,b}$. The different constraints that correspond to Carroll 2, 3 and 4 geometry have a clear geometric interpretation. For instance, the Carroll 2 constraint implies that the $(D-1)$-form $\epsilon_{a_1 \cdots a_{D-1}} e_{[\mu_1}{}^{a_1} \cdots e_{\mu_{D-1}]}{}^{a_{D-1}}$ is closed and this closure can be physically interpreted as stating that there exists a notion of absolute spatial volumes. The Carroll 3 constraint can be rephrased as saying that $\tau^\mu$ is a conformal Killing vector with respect to the spatial metric $h_{\mu\nu} = e_\mu{}^a e_{\nu a}$. We refer to [38] for further details.

While the number of Carrollian first Cartan structure equations (59) equals the number of Carrollian spin-connection components, the presence of intrinsic torsion components implies that not all of them can be used to solve for components of $\omega_\mu{}^{ab}$ and $\omega_\mu{}^{0a}$. Not all spin-connection components can thus be expressed in terms of the Carrollian solder form, Vielbein and torsion tensors $t_{\mu\nu}{}^0$, $t_{\mu\nu}{}^a$. Explicitly, one finds that the components

$$\omega^{(a|0|b)} \equiv e^{(a|\mu} \omega_\mu{}^{0|b)}, \tag{61}$$

can not be solved from the first Cartan structure equations (59). We will refer to them as the *independent* spin-connection components. The remaining components will be called *dependent* and by solving them from (59) one finds that they can be expressed in terms of the Carrollian solder form, Vielbein and torsion tensors as follows[21]

$$\omega_0{}^{ab} = \omega_0{}^{ab}(\tau, e) - t_0{}^{[a,b]}, \qquad \omega_c{}^{ab} = \omega_c{}^{ab}(e) - t_c{}^{[a,b]} + \frac{1}{2} t^{ab}{}_c,$$

$$\omega_0{}^{0a} = \omega_0{}^{0a}(\tau, e) + t_0{}^{a,0}, \qquad \omega^{[a|0|b]} = \omega^{[a|0|b]}(\tau, e) + \frac{1}{2} t^{ab,0}, \tag{62}$$

with

$$\omega_0{}^{ab}(\tau, e) = e_0{}^{[a,b]}, \qquad \omega_c{}^{ab}(e) = e_c{}^{[a,b]} - \frac{1}{2} e^{ab}{}_c,$$

$$\omega_0{}^{0a}(\tau, e) = -\tau_0{}^a, \qquad \omega^{[a|0|b]}(\tau, e) = -\frac{1}{2} \tau^{ab}. \tag{63}$$

---

[21]To prevent confusion, we use a notation in which we put a comma between the indices to indicate which two indices form (the projection of) an anti-symmetric pair, e.g., we write $t_0{}^{a,0} = \tau^\mu e^{a\nu} t_{\mu\nu}{}^0$ to clarify that the first two indices (0 and $a$) are those of an anti-symmetric pair.

The appearance of intrinsic torsion and independent spin-connection components is not restricted to Carroll geometry but is a generic feature of non-Lorentzian geometry and has been studied in full generality in [38].

Let us end this section by recalling that the Lorentzian first Cartan structure equations (53) for zero torsion ($T_{\mu\nu}{}^A = 0$) correspond to the equations of motion of $\Omega_\mu{}^{AB}$ of the Einstein-Hilbert action in the first-order formulation

$$S_{\text{EH}} = \frac{1}{16\pi G_N} \int \mathrm{d}^D x \, E E_A{}^\mu E_B{}^\nu R_{\mu\nu}{}^{AB}, \quad \text{with} \quad R_{\mu\nu}{}^{AB} = 2\,\partial_{[\mu}\Omega_{\nu]}{}^{AB} + 2\,\Omega_{[\mu}{}^A{}_C \Omega_{\nu]}{}^{CB}. \quad (64)$$

Likewise, their zero torsion ($t_{\mu\nu}{}^0 = 0 = t_{\mu\nu}{}^a$) Carrollian analogue (59) can be derived as equations of motion for $\omega_\mu{}^{ab}$ and $\omega_\mu{}^{0a}$ of the following first-order 'magnetic Carroll gravity' action [37, 41, 42]:

$$S_{\text{Carr. Grav.}} = \frac{1}{16\pi G_C} \int \mathrm{d}^D x \, e\Big(e_a{}^\mu e_b{}^\nu R_{\mu\nu}(J)^{ab} + 2\tau^\mu e_a{}^\nu R_{\mu\nu}(C)^{0a}\Big), \quad (65)$$

$$\text{with} \quad R_{\mu\nu}(J)^{ab} = 2\,\partial_{[\mu}\omega_{\nu]}{}^{ab} + 2\,\omega_{[\mu}{}^a{}_c \,\omega_{\nu]}{}^{cb},$$

$$\text{and} \quad R_{\mu\nu}(C)^{0a} = 2\,\partial_{[\mu}\omega_{\nu]}{}^{0a} + 2\,\omega_{[\mu}{}^{ab} \,\omega_{\nu]}{}^0{}_b.$$

This action is obtained by using the rescalings (55) (along with $G_N = G_C/\tilde{c}$) in (64) and taking the $\tilde{c} \to \infty$ limit. It is interesting to note that the spin-connection components $\omega^{(a|0|b)}$ only appear linearly in it and that they assume the role of Lagrange multipliers for the Carroll 4 constraints [37, 38, 42]. Variation of (65) with respect to $\omega_\mu{}^{ab}$ and $\omega_\mu^{0a}$ leads to

$$\delta S_{\text{Carr. Grav.}} = \frac{1}{8\pi G_C} \int \mathrm{d}^D x \, e\bigg[\delta\omega_\mu{}^{ab}\Big(-e_a{}^\mu t_{0b}{}^0 + e_a{}^\mu t_{bc}{}^c + \frac{1}{2}e_c{}^\mu t_{ab}{}^c + \frac{1}{2}\tau^\mu t_{ab}{}^0\Big)$$

$$+ \delta\omega_\mu{}^{0a}\Big(-e_a{}^\mu t_{0b}{}^b + e_b{}^\mu t_{0a}{}^b + \tau^\mu t_{ab}{}^b\Big)\bigg]. \quad (66)$$

From this, one sees that the Carrollian first Cartan structure equations (59) for zero torsion are derived from (65) as the equations of motion of the spin-connections.

In the next section, we will discuss the coupling of both electric and magnetic Carroll fermions to this first-order Carroll gravity action.

## 3.2 Coupling Carroll fermions to gravity in the first-order formulation

In General Relativity, fermionic matter can be coupled to the Einstein-Hilbert action in the first-order formulation, by adding a Dirac Lagrangian suitably coupled to the Vielbein and first-order spin-connection $\Omega_\mu{}^{AB}$:

$$S = S_{\text{EH}} + \int \mathrm{d}^D x \, E\bigg[-\frac{1}{2}\bar{\Psi}E_A{}^\mu \Gamma^A\Big(\partial_\mu\Psi + \frac{1}{4}\Omega_\mu{}^{BC}\Gamma_{BC}\Psi\Big) + \frac{M}{2\tilde{c}}\bar{\Psi}\Psi + \text{h.c.}\bigg]. \quad (67)$$

The first Cartan structure equations that result as equations of motion of $\Omega_\mu{}^{AB}$ then receive torsion contributions in the form of fermion bilinears. When passing to the second-order formulation, by replacing the spin-connections $\Omega_\mu{}^{AB}$ in (67) by their dependent expressions (54), these fermion bilinear torsion components give rise to an action that includes quartic fermion terms. In this section, we will investigate the analogous coupling of Carroll fermions to magnetic Carroll gravity (65) in the first-order formulation and we will comment on the passage to the second-order formulation. We will discuss the cases of electric and magnetic Carroll fermions in turn.

Let us first focus on the electric case. By applying the rescalings (55) (along with $\Psi = \psi$ and $M = m\tilde{c}^2$) to (67) and taking the $\tilde{c} \to \infty$ limit, we obtain the action of an electric Carroll fermion, coupled to magnetic Carroll gravity in the first-order formalism:[22]

$$S = S_{\text{Carr. Grav.}} + \int \mathrm{d}^D x\, e \left[ -\frac{1}{2} \bar{\psi} \Gamma^0 \tau^\mu D_\mu \psi + \frac{m}{2} \bar{\psi} \psi + \text{h.c.} \right], \tag{68}$$

$$\text{with } D_\mu \psi = \partial_\mu \psi + \frac{1}{4} \omega_\mu{}^{ab} \Gamma_{ab} \psi.$$

The equations of motion of $\omega_\mu{}^{ab}$ and $\omega_\mu{}^{0a}$ that follow from (68) correspond to the Carrollian first Cartan structure equations (59) with

$$t_{0a}{}^0 = 0, \qquad t_{ab}{}^0 = \frac{\kappa_C^2}{2} \bar{\psi} \Gamma^0 \Gamma_{ab} \psi, \qquad t_{\mu\nu}{}^a = 0, \quad \text{with} \quad \kappa_C^2 = 8\pi G_C. \tag{69}$$

The first-order coupling of an electric Carroll fermion to General Relativity thus introduces bilinear fermion torsion.

To pass to the second-order formulation, one replaces the dependent spin-connection components $\omega_\mu{}^{ab}$, $\tau^\mu \omega_\mu{}^{0a}$ and $e^{[a|\mu} \omega_\mu{}^{0|b]}$ in the action (68) by their explicit expressions (62) (with the torsion tensor components replaced by (69)). This leads, after partial integration, to the following action

$$S = \int \mathrm{d}^D x\, e \left[ \frac{1}{2\kappa_C^2} \left( \omega_b{}^{ab}(e) \omega_{ca}{}^c(e) - \omega_{abc}(e) \omega^{bac}(e) - 2\omega_0{}^{0a}(\tau, e) \omega_{ba}{}^b(e) \right. \right.$$
$$\left. - 2\omega_{[a|0|b]}(\tau, e) \omega_0{}^{ab}(\tau, e) + 2\omega^{(a|0|b)} e_{0(a,b)} - 2\omega_a{}^{0a} e_{0a}{}^a \right)$$
$$\left. + \left\{ -\frac{1}{2} \bar{\psi} \Gamma^0 \tilde{D}_0 \psi + \frac{m}{2} \bar{\psi} \psi + \text{h.c.} \right\} \right], \tag{70}$$

with

$$\tilde{D}_0 \psi \equiv \tau^\mu \partial_\mu \psi + \frac{1}{4} \omega_0{}^{ab}(\tau, e) \Gamma_{ab} \psi. \tag{71}$$

Note that the bilinear torsion (69) does not lead to quartic fermion terms, when passing to the second-order formulation, in contrast to what happens for a Dirac fermion coupled to General Relativity. The same observation was reached in [32] while linking the first-order formulation of Carroll gravity coupled to Dirac fermions to its Hamiltonian formulation.

To discuss the coupling of magnetic Carroll fermions to Carroll gravity, we assume that the spacetime dimension $D$ is even (and $> 2$) and start from the second of the two actions (33) for a tachyonic Dirac spinor, coupled to the Einstein-Hilbert action in the first-order formulation:

$$S = S_{\text{EH}} + \int \mathrm{d}^D x\, E \left[ -\frac{1}{2} \bar{\Psi} E_A{}^\mu \Gamma^A \Gamma_\star \left( \partial_\mu \Psi + \frac{1}{4} \Omega_\mu{}^{BC} \Gamma_{BC} \Psi \right) + \frac{M}{2\tilde{c}} \bar{\Psi} \Psi + \text{h.c.} \right]. \tag{72}$$

By splitting $\Psi$ into $\Psi_\pm$, using the rescalings (55) (along with $\Psi_+ = \tilde{c}^{1/2} \psi_+$, $\Psi_- = \tilde{c}^{-1/2} \psi_-$ and $M = \tilde{c}\, m$) to (72) and taking the $\tilde{c} \to \infty$ limit, the following action for a minimal magnetic Carroll fermion coupled to magnetic Carroll gravity in the first-order formalism is obtained:

$$S = S_{\text{Carr. Grav.}} + \int \mathrm{d}^D x\, e \left[ -\frac{1}{2} \bar{\psi}_+ \Gamma^0 \Gamma_\star \tau^\mu D_\mu \psi_- - \frac{1}{2} \bar{\psi}_- \Gamma^0 \Gamma_\star \tau^\mu D_\mu \psi_+ \right.$$
$$\left. - \frac{1}{2} \bar{\psi}_+ \Gamma^a \Gamma_\star e_a{}^\mu D_\mu \psi_+ + \frac{m}{2} \bar{\psi}_+ \psi_+ + \text{h.c.} \right], \tag{73}$$

---

[22]This action was already considered in [37].

where

$$D_\mu \psi_+ \equiv \partial_\mu \psi_+ + \frac{1}{4}\omega_\mu{}^{ab}\Gamma_{ab}\psi_+, \qquad D_\mu \psi_- \equiv \partial_\mu \psi_- + \frac{1}{4}\omega_\mu{}^{ab}\Gamma_{ab}\psi_- + \frac{1}{2}\omega_\mu{}^{0a}\Gamma_{0a}\psi_+. \quad (74)$$

In this case, the torsion tensor components of the Carrollian first Cartan structure equations (59) that arise as equations of motion of $\omega_\mu{}^{ab}$ and $\omega_\mu{}^{0a}$ are given by

$$t_{0a}{}^0 = 0, \qquad t_{ab}{}^0 = \frac{\kappa_C^2}{2}\bar{\psi}_+\Gamma^0\Gamma_{ab}\Gamma_\star\psi_- + \frac{\kappa_C^2}{2}\bar{\psi}_-\Gamma^0\Gamma_{ab}\Gamma_\star\psi_+,$$

$$t_{0b}{}^a = 0, \qquad t_{bc}{}^a = \frac{\kappa_C^2}{2}\bar{\psi}_+\Gamma^a{}_{bc}\Gamma_\star\psi_+. \quad (75)$$

Going to the second-order formulation, by replacing $\omega_\mu{}^{ab}$, $\tau^\mu\omega_\mu{}^{0a}$ and $e^{[a|\mu}\omega_\mu{}^{0|b]}$ in (73) by their expressions (62) in terms of the Carroll solder form, Vielbein and torsion tensor components (75) now leads (after partial integration) to the following action:

$$S = \int \mathrm{d}^4 x\, e\left[ \frac{1}{2\kappa_C^2}\left( \omega_b{}^{ab}(e)\omega_{ca}{}^c(e) - \omega_{abc}(e)\omega^{bac}(e) - 2\omega_0{}^{0a}(\tau,e)\omega_{ba}{}^b(e) \right.\right.$$

$$\left. -2\omega_{[a|0|b]}(\tau,e)\omega_0{}^{ab}(\tau,e) + 2\omega^{(a|0|b)}e_{0(a,b)} - 2\omega_a{}^{0a}e_{0a}{}^a \right)$$

$$-\frac{1}{2}\left\{ \bar{\psi}_+\Gamma^0\Gamma_\star\tilde{D}_0\psi_- + \bar{\psi}_-\Gamma^0\Gamma_\star\tilde{D}_0\psi_+ + \bar{\psi}_+\Gamma^a\Gamma_\star\tilde{D}_a\psi_+ + \text{h.c.} \right\}$$

$$\left. + m\bar{\psi}_+\psi_+ + \frac{\kappa_C^2}{32}\left(\bar{\psi}_+\Gamma^{abc}\Gamma_\star\psi_+\right)\left(\bar{\psi}_+\Gamma_{abc}\Gamma_\star\psi_+\right) \right], \quad (76)$$

with

$$\tilde{D}_0\psi_+ \equiv \tau^\mu\partial_\mu\psi_+ + \frac{1}{4}\omega_0{}^{ab}(\tau,e)\Gamma_{ab}\psi_+, \qquad \tilde{D}_a\psi_+ \equiv e_a{}^\mu\partial_\mu\psi_+ + \frac{1}{4}\omega_a{}^{bc}(e)\Gamma_{bc}\psi_+,$$

$$\tilde{D}_0\psi_- \equiv \tau^\mu\partial_\mu\psi_- + \frac{1}{4}\omega_0{}^{ab}(\tau,e)\Gamma_{ab}\psi_- + \frac{1}{2}\omega_0{}^{0a}(\tau,e)\Gamma_{0a}\psi_+. \quad (77)$$

Unlike what happened for the electric Carroll fermion, the bilinear fermion torsion (75) does lead to quartic fermion terms when one writes the first-order action (73) in second-order form.

For both the electric and magnetic Carroll fermion, the spatial parts $e_b{}^\mu\omega_\mu{}^{0a}$ of the boost spin-connection is not involved in the coupling to gravity. From (66), one then sees that adding each of these first-order fermion Lagrangians to (65) does not lead to sources for the $t_{0a}{}^b$ components of the torsion. The resulting system thus still obeys the Carroll 4 geometry constraints, for which all intrinsic torsion components $t_{0(a,b)}$ are equal to zero. The remark made below (65), namely that the $\omega^{(a|0|b)}$ spin-connection components are Lagrange multipliers for the Carroll 4 constraints, is thus unaffected by the coupling to fermions and this is clearly visible in the second order Lagrangians (70) and (76).

### 3.3 Carroll limit of fermions coupled to gravity in the second-order formulation

In the previous section, we obtained first-order actions for an electric and magnetic Carroll fermion, coupled to Carroll gravity, by taking a Carroll limit of (67) and (72). This limit can also be taken directly in the second-order formulation in which the spin-connection $\Omega_\mu{}^{AB}$ is no longer independent, but instead given by the Levi-Civita spin-connection $\Omega_\mu{}^{AB}(E)$:

$$\Omega_\mu{}^{AB}(E) = 2E^{[A|\nu|}\partial_{[\mu}E_{\nu]}{}^{B]} - E_{\mu C}E^{A\nu}E^{B\rho}\partial_{[\nu}E_{\rho]}{}^C. \quad (78)$$

Upon applying the rescalings of $E_\mu{}^A/E_A{}^\mu$ given in (55), the components of $\Omega_C{}^{AB}(E) = E_C{}^\mu \Omega_\mu{}^{AB}(E)$ can be expanded in powers of $\tilde{c}$ as follows:

$$\Omega_0{}^{ab}(E) = \tilde{c}\,\omega_0{}^{ab}(\tau, e) + \mathcal{O}\left(\tilde{c}^{-1}\right), \qquad \Omega_c{}^{ab}(E) = \omega_c{}^{ab}(e),$$

$$\Omega_0{}^{0a}(E) = \omega_0{}^{0a}(\tau, e), \qquad \Omega^{a,0b}(E) = \tilde{c}\,t^{0(a,b)} + \tilde{c}^{-1}\,\omega^{[a|0|b]}(\tau, e). \quad (79)$$

Note that the $\tilde{c}$-power of the leading-order term in the expansions of $\Omega_0{}^{ab}(E)$, $\Omega_c{}^{ab}(E)$ and $\Omega_0{}^{0a}(E)$ agrees with that of the rescalings (55) of the corresponding spin-connection components $\Omega_C{}^{AB} = E_C{}^\mu \Omega_\mu{}^{AB}$ in the first-order formalism. This is however not true for $\Omega^{a,0b}(E)$, for which the $\tilde{c}$-power suggested by the rescalings (55) only appears at subleading order. Compared to the rescaling of $\Omega^{a,0b}$ in the first-order formulation, the $\tilde{c}$-expansion of $\Omega^{a,0b}(E)$ contains an additional 'divergent' $\mathcal{O}(\tilde{c})$-contribution that is given by the intrinsic torsion components (60).

Let us then start from (67) and (72) in the second-order formulation (i.e., with $\Omega_\mu{}^{AB}$ replaced by $\Omega_\mu{}^{AB}(E)$). After setting $G_N = G_C/\tilde{c}$, rescaling $E_\mu{}^A/E_A{}^\mu$ as in (55) and $\Psi$ and $M$ as in the electric, resp. magnetic case of the previous subsection, the $\tilde{c} \to \infty$ limit of (67), resp. (72) is given by the leading order term in their $\tilde{c}$-expansion. These leading order terms are of order $\mathcal{O}(\tilde{c}^2)$, instead of order $\mathcal{O}(\tilde{c}^0)$ as was the case for the analogous expansions in the first-order formulation, due to the divergent $\mathcal{O}(\tilde{c})$-contribution in the expansion (79) of $\Omega^{a,0b}(E)$. In particular, one gets for both (67) and (72) the following expansion:

$$S \propto \tilde{c}^2 \int \mathrm{d}^D x\, e \left( t_0{}^{\{a,b\}} t_{0\{a,b\}} - \frac{D-2}{D-1} t_{0a}{}^a t_{0b}{}^b \right) + \mathcal{O}(\tilde{c}^0). \quad (80)$$

One thus sees that no fermion terms survive the $\tilde{c} \to \infty$ limit, whose result is the so-called 'electric Carroll gravity' theory [43] (see also [12, 44, 45]):

$$S_{\text{el. Carr. Grav.}} = \frac{1}{16\pi G_C} \int \mathrm{d}^D x\, e \left( t_0{}^{\{a,b\}} t_{0\{a,b\}} - \frac{D-2}{D-1} t_{0a}{}^a t_{0b}{}^b \right). \quad (81)$$

It is possible to take a Carroll limit in the second-order formalism that results in theories that include fermion contributions and that closely resemble (70) and (76). To do this, one applies a Hubbard-Stratonovich transformation to the leading order term of (80), i.e., one introduces auxiliary fields $\chi^{ab} = \chi^{\{ab\}}$ and $\chi$ to rewrite this term in the classically equivalent form

$$\int \mathrm{d}^D x\, e \left( \tilde{c}^\alpha \chi^{ab} t_{0\{a,b\}} - \tilde{c}^\beta \frac{D-2}{D-1} \chi t_{0a}{}^a - \frac{\tilde{c}^{2\alpha-2}}{4} \chi^{\{ab\}} \chi_{\{ab\}} + \frac{\tilde{c}^{2\beta-2}}{4} \frac{D-2}{D-1} \chi^2 \right), \quad (82)$$

where $\alpha, \beta \leq 0$. Applying this transformation with the choice $\alpha = 0 = \beta$, one finds that the resulting action has the following term at leading order:

$$S = \int \mathrm{d}^D x\, e \left[ \frac{1}{2\kappa_C^2} \left( \omega_b{}^{ab}(e) \omega_{ca}{}^c(e) - \omega_{abc}(e) \omega^{bac}(e) - 2\omega_0{}^{0a}(\tau, e) \omega_{ba}{}^b(e) \right. \right.$$

$$\left. - 2\omega^{[a|0|b]}(\tau, e) \omega_0{}^{ab}(\tau, e) + \chi^{ab} e_{0\{a,b\}} - \frac{D-2}{D-1} \chi e_{0a}{}^a \right)$$

$$\left. - \frac{1}{2} \left\{ \bar{\psi} \Gamma^0 \tilde{D}_0 \psi + \text{h.c.} \right\} + m \bar{\psi} \psi \right]. \quad (83)$$

For the action (72), one gets:

$$S = \int \mathrm{d}^D x\, e \left[ \frac{1}{2\kappa_C^2} \left( \omega_b{}^{ab}(e) \omega_{ca}{}^c(e) - \omega_{abc}(e) \omega^{bac}(e) - 2\omega_0{}^{0a}(\tau, e) \omega_{ba}{}^b(e) \right. \right.$$

$$\left. - 2\omega^{[a|0|b]}(\tau, e) \omega_0{}^{ab}(\tau, e) + \chi^{ab} e_{0\{a,b\}} - \frac{D-2}{D-1} \chi e_{0a}{}^a \right)$$

$$\left. - \frac{1}{2} \left\{ \bar{\psi}_+ \Gamma^0 \Gamma_\star \tilde{D}_0 \psi_- + \bar{\psi}_- \Gamma^0 \Gamma_\star \tilde{D}_0 \psi_+ + \bar{\psi}_+ \Gamma^a \Gamma_\star \tilde{D}_a \psi_+ + \text{h.c.} \right\} + m \bar{\psi}_+ \psi_+ \right]. \quad (84)$$

Upon identifying $\chi^{ab}$ and $\chi$ with $2\omega^{\{a|0|b\}}$ and $2\omega_a{}^{0a}$, one sees that this way of taking the limit reproduces the results (70) and (76) of the limit of the first-order formulation, apart from the quartic fermion terms in (76).

One can also apply the Hubbard-Stratonovich transformation (82) with different values of $\alpha$ and $\beta$. Taking $\alpha = 0$ and e.g., $\beta = -2$, leads to the actions (83) and (84) without the $\chi e_{0a}{}^a$ term. Only the auxiliary field $\chi^{\{ab\}}$ is retained in the limit and it acts as a Lagrange multiplier for the Carroll 3 constraint. Likewise, taking $\beta = 0$ and e.g., $\alpha = -2$ leads to the actions (83) and (84) without the $\chi^{ab}e_{0\{a,b\}}$ term and the remaining field $\chi$ plays the role of Lagrange multiplier for the Carroll 2 constraint. Finally, in the above scenarios, we applied the Hubbard-Stratonovich transformation to both leading order terms of (80). By using it more selectively, one can obtain two electric Carroll gravity theories, other than the one of (81). In particular, applying it to only the $t_0{}^{\{a,b\}}t_{0\{a,b\}}$ leading order term of (80) leads in the $\tilde{c} \to \infty$ limit to an electric Carroll gravity action proportional to

$$\int \mathrm{d}^D x \, e \, t_{0a}{}^a t_{0b}{}^b \,. \tag{85}$$

Likewise, an action proportional to

$$\int \mathrm{d}^D x \, e \, t_0{}^{\{a,b\}} t_{0\{a,b\}} \tag{86}$$

is obtained by applying the Hubbard-Stratonovich trick to only the $t_{0a}{}^a t_{0b}{}^b$ leading order contribution of (80). One can even apply a further Hubbard-Stratonovich transformation to each of these, to obtain actions proportional to

$$\int \mathrm{d}^D x \, e \, \chi \, t_{0a}{}^a \,, \qquad \int \mathrm{d}^D x \, e \, \chi^{\{ab\}} t_{0\{a,b\}} \,, \tag{87}$$

that just consist of Lagrange multiplier terms that impose the Carroll 2 and Carroll 3 constraints respectively.

## 4 Conclusions

In this work we showed how to obtain electric and magnetic Carroll fermions by taking Carroll limits of relativistic Dirac fermions. We first studied the possible transformations of spinor fields under the homogeneous Carroll group and then we identified dynamical models implementing the various options. Similarly to the bosonic case, the electric theory can be defined simply by considering the limit of vanishing speed of light of the Dirac action. Magnetic limits are instead subtler: in odd spacetime dimensions they require to double the number of spinors in the relativistic theory. Due to this subtlety, and in order to allow for easier comparisons with the existing literature, we derived our magnetic actions in two complementary ways. We first started from an off-diagonal two-fermion action and then we rederived the same result starting from a singular limit of the Dirac action rewritten in Hamiltonian form. In the second approach the doubling of spinorial variables is induced by the loss of second class constraints in the limit. We however stressed that in even dimensions a truncation of the magnetic theory is possible, leading to a magnetic Carroll fermion with the same number of components as a single Dirac fermion. In this case, one can also avoid the doubling of fermionic variables to begin with and define the Carroll limit starting from the tachyonic action (33).

Similar limits of General Relativity have been considered before leading to electric [43] and magnetic [12, 37, 41, 42] Carroll gravity theories. By combining the two limits we obtained

expressions for electric and magnetic Carroll fermions coupled to magnetic Carroll gravity. It would be interesting to see whether, using different techniques, electric Carrol gravity can also be coupled to fermions.

We note that the results on Carroll limits can be generalised to similar Galilei limits of both scalars [46], fermions [24] and gravity [37]. This leads to similar notions such as electric and magnetic Galilei fermions and electric and magnetic Galilei gravity. For instance, the expression for electric Galilei gravity is given by the expression

$$S_{\text{el. Galilei. grav.}} = \frac{1}{16\pi G_G} \int \mathrm{d}^4 x\, e\, t^{ab} t_{ab}\,, \tag{88}$$

where $t_{ab} = e_a{}^\mu e_b{}^\nu (\partial_\mu \tau_\nu - \partial_\nu \tau_\mu)$ and $\tau_\mu$ is the clock function.

An undesirable feature of our limit technique is that it requires the existence of a Dirac fermion. This excludes for instance applications to supergravity in ten and eleven spacetime dimensions. This obstacle can be circumvented by generalising our particle limit to extended objects where the $\Gamma_0$ occurring in the projection operator is replaced by a gamma matrix with indices in all directions longitudinal to the extended object [47]. For instance, in ten and eleven dimensions where the corresponding supergravity theories do not contain Dirac spinors one could define string and membrane projection operators as follows:

$$P_\pm = \frac{1}{2}(1 \pm \Gamma_{01}) : \text{strings}, \quad \text{and} \quad P_\pm = \frac{1}{2}(1 \pm \Gamma_{012}) : \text{membranes}. \tag{89}$$

These projection operators are consistent with Majorana-Weyl spinors in 10D and Majorana spinors in 11D.

An advantage of our limit is that the rescalings with $\tilde{c}$ guarantee that our final expressions are invariant under global scalings of the fields. This may imply that the Carroll symmetry is extended to a conformal Carroll symmetry. This leads to interesting connections with the BMS symmetry that plays such a prominent role in flat space holography and celestial holography. In this context, it is of interest to note that the free Carroll fermion models we constructed may be used to construct infinite dimensional algebras of the type $w_{1+\infty}$ in the same way as this has been done before in the relativistic case, see e.g. [48,49]. It would be interesting to see whether there exist electric and magnetic versions of these infinite-dimensional algebras.

Last but not least, given that we have electric (magnetic) Carroll scalars and fermions, it is natural to consider supersymmetric combinations. We hope to report on this in a future work [50] (see also [15, 24, 25]).

## Acknowledgments

We thank Nicolas Boulanger, Dario Francia, Laurent Freidel, Marios P. Petropoulos for discussions.

AF thanks Lia for her permanent support. EB thanks Perimeter Institute for hospitality. AC, EB and JR thank the organisers of the workshops "Strings, Branes and Gravitational Waves" (Solvay Institutes, Brussels, 22/8/2023), "Quantum Gravity, Strings and the Swampland" (Corfu, 12–19/9/2023) and of the "3rd Carroll workshop" (Aristotle University of Thessaloniki, 2–6/10/2023) for hospitality and for the opportunity to present a preliminary version of the results of this paper.

**Funding information** AC is a research associate of the Fonds de la Recherche Scientifique – FNRS. The work of AC and LM was supported in part by FNRS under Grants FC.55077, F.4503.20 and T.0047.24. This research was supported in part by Perimeter Institute for Theoretical Physics. Research at Perimeter Institute is supported in part by the Government of Canada through the Department of Innovation, Science and Economic Development and by the Province of Ontario through the Ministry of Colleges and Universities. During the last part of this work, AF was supported by SFI and the Royal Society under the grant number RFF\EREF\210373.

# A Conventions

We use capital Latin indices $A$, $B$, $\cdots (= 0, 1, \cdots, D-1)$ to denote inertial Minkowski coordinates. These indices will often be split in a time-like one, denoted by 0, and spatial ones, denoted by lowercase Latin indices $a$, $b$, $\cdots (= 1, \cdots, D-1)$. The Minkowski metric $\eta_{AB}$ used in this paper has mostly plus signature: $\eta_{AB} = \mathrm{diag}(-1, +1, +1, \cdots, +1)$.

We take the $\Gamma$-matrices to obey the following Clifford algebra relation:

$$\{\Gamma_A, \Gamma_B\} = 2\eta_{AB}\mathbb{1}\,. \tag{A.1}$$

They satisfy the following Hermiticity properties:

$$\left(\Gamma^0\right)^\dagger = -\Gamma^0\,, \qquad (\Gamma^a)^\dagger = \Gamma^a \qquad (a = 1, 2, \cdots, D-1)\,. \tag{A.2}$$

We define the matrix $\Gamma_\star$ in terms of the product of all $\Gamma$-matrices as:

$$\Gamma_\star \equiv (-\mathrm{i})^{\frac{D}{2}+1}\Gamma^0\Gamma^1\cdots\Gamma^{D-1}\,. \tag{A.3}$$

It obeys the following properties:

$$\Gamma_\star^\dagger = \Gamma_\star\,, \qquad \Gamma_\star^2 = 1\,. \tag{A.4}$$

In $D = 4$, $\Gamma_* = \mathrm{i}\Gamma^0\Gamma^1\Gamma^2\Gamma^3$ corresponds to the matrix that is usually called $\Gamma_5$.

The Dirac conjugate of a spinor is defined as

$$\bar{\Psi} \equiv \mathrm{i}\Psi^\dagger\Gamma^0\,. \tag{A.5}$$

The complex conjugate of a product of (Grassmann-valued) spinor components $\varepsilon_\alpha$, $\psi_\beta$ is defined as the product of their complex conjugates in reverse order:

$$(\varepsilon_\alpha \psi_\beta)^* = \psi_\beta^* \varepsilon_\alpha^*\,. \tag{A.6}$$

We note the following useful commutation properties of the projectors $P_\pm$ introduced in (11)

$$P_\pm \Gamma^0 = \Gamma^0 P_\pm\,, \qquad P_\pm \Gamma^a = \Gamma^a P_\mp\,, \qquad P_\pm \Gamma_\star = \Gamma_\star P_\mp\,. \tag{A.7}$$

Furthermore, projected spinors obey the following properties:

$$\bar{\Psi}_\pm = \bar{\Psi}_\pm P_\pm\,, \qquad \Gamma^0\Psi_\pm = \mp\mathrm{i}\Psi_\pm\,. \tag{A.8}$$

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
