# Peer review of "Carroll Fermions"

_SciPost Physics, doi:SciPost Phys. 16, 153 (2024)_

## Round 1 · Referee Report · Stefan Vandoren (Referee 1) · 2024-1-17

Strengths

  1. Finally a systematic and clear analysis on the topic of Carroll fermions.
  2. Well written paper, solid and elegant analysis.
  3. The new results found in this paper will be useful for many upcoming applications.

Weaknesses

  1. While the results are expected to have many applications, no single application is given. Instead, the paper focusses on the construction of Lagrangians and couplings to gravity rather than on what to do with such models.

Report

This paper shows significant progress on the construction of actions for Carroll fermions and their couplings to gravity. This topic is addressed before in the literature, but often in an unclear or un-elegant way, or in too specific cases. The results presented in this paper hold in arbitrary dimension, and with an elegant treatment of Dirac-gamma matrices in the Carroll limit. The techniques are solid, some of the authors have made important contributions in this field before and are experts. The presentation is very good and references are given to the literature in an appropriate way. I recommend publication.

---

## Round 1 · Referee Report · Anonymous (Referee 2) · 2024-3-27

Strengths

  1. Well-written and easily understandable paper.
  2. Technical details are also straight forward.

Weaknesses

  1. Slight lack of explanation at some points.
  2. No practical reason for understanding Carroll fermions have been mentioned.

Report

I could recommend the publication of the article “Carroll Fermions” by E.A.Bergshoeff et. Al. after some minor clarifications.

My doubts are the following: 1. Starting from eq.(2.1) and after rescaling the spacetime coordinates and Lorentz transformation parameters, the authors found eq.(2.5) and they claimed "only spatial rotations appear in the spin part of the transformation rule of the above Carroll fermion." But in the transformation eq.(2.1) there is also a factor $\Gamma_{AB}$ which I guess is the combination of gamma matrices. What is the reason behind not scaling this factor? Will it remain unaffected after taking Carroll limit? A similar doubt can be asked for the case of eq.(2.9).

2. In eq.(2.6) the authors have defined the Projection operator which satisfies the mentioned properties. As far as I know the projection operator is in general defined in terms of $\Gamma_*$ matrices instead of $\Gamma^0$. Is there any specific reason for choosing such definition?

3. The authors have written the electric Carroll Lagrangian in eq.(2.17) following the truncation eq.(2.16). For electric Carroll Lagrangian only $\psi_{+}$ and its Hermitian conjugate appear in the Lagrangian. It seems the degrees of freedom somehow got reduced compared to the relativistic fermions. Is it an artefact of the scaling procedure or the property of the electric Carroll theory? In the following section, the authors have mentioned the same action can be obtained by taking $\psi_- = 0$. Other than matching the actions what is the reason behind it?

4.  Although the authors haven't mentioned anything about representations of $\Gamma$ matrices, I am curious about this. For the relativistic case, we know every representation of Cliff$(1,d)$ induces a representation of $so(1,d)$. Does in the Carroll case we get something like a contraction of $so(1,d)$?

5.  As can be seen the mass term for electric (eq.(2.17)) and magnetic (eq.(2.23)) Carroll fermions are different. For the relativistic case, among the invariant bilinears $\bar{\psi}\psi$ and $\bar{\psi}\Gamma_*\psi$ are scalar and psedo-scalar respectively and accordingly one can get parity-even or parity-odd Lagrangian. Is it the same here?

---

## Round 2 · Referee Report · Anonymous (Referee 2) · 2024-4-30

Strengths
- Well-written and easily understandable paper.
- Technical details are also straight forward.
Weaknesses
- No practical reason for understanding Carroll fermions have been mentioned.
Report
Dear Editor,
The resubmission of $\textit{Carroll Fermions}$ by E.A.Bergshoeff, A.Campoleoni, A.Fontanella, L.Mele and J.Rosseel addresses most of the points raised in my first review. However, a few clarifications are still required.
-
As shown in the paper arXiv: 2109.06708, for any Carroll invariant theory, the energy density $\mathcal{E}(x)$ must satisfy the following property: $[\mathcal{E}(x),\mathcal{E}(x')]=0$. I can see this is obvious for the electric Carroll fermion. But how does this hold for magnetic ones?
-
As the authors pointed out, there is nothing special with $\psi_+$ in the electric limit, one can also do the same with $\psi_-$. But my question was, as both together can't be present in the electric Lagrangian, then it's obvious that degrees of freedom are getting halved. What is the physical reason behind this?
-
The authors have mentioned "...the purpose of having introduced the parent Lagrangian (2.10) is to unify the electric limit with the magnetic limit discussed below". However if one performs c-expansion from relativistic theory, electric and magnetic theory appear at different order, the latter comes with additional constraints as done in the paper arXiv: 2110.02319. So how can one unify the limits ?
Recommendation
Ask for minor revision

---

## Round 2 · Author Response

First of all we wish to thank the Referees for the useful comments. We added some clarifications according to the suggestions by Referee 2, that we detail in the following list of changes. We hope that these additional clarifications will make our paper suitable for publication.

---

## Round 2 · List of Changes

-
Below (2.2) we defined $\Gamma_{AB}$ and below (2.3) we stressed that in our approach we do not rescale gamma matrices.
-
Below (2.6) we clarified that we used $\Gamma_0$ to define a projection operator because our aim was to introduce a projection allowing us to distinguish between rotations and boosts. This was motivated by the desire to rescale fields in such a way to preserve a non-trivial action of boosts. We also modified the paragraph below (2.9) to recall how we obtained an indecomposable representation of the homogeneous Carroll group even if we worked with the usual relativistic Carroll algebra.
-
The electric limit can be actually defined in a simpler way by taking the $\tilde{c} \to \infty$ limit of Dirac's Lagrangian after rescaling the fields by as $\psi = \tilde{c}^{-\frac{1}{2}} \Psi$. We stressed at the beginning of section 2.2 that our choice of starting from an off-diagonal Lagrangian was motivated by the desire to describe both an electric and a magnetic limit in a unifying framework. We also stressed in footnote 10 and in the paragraph below (2.17) that our choice of keeping only the $+$ components in the electric limit is purely conventional and that the truncation (with the corresponding reduction in the number of degrees of freedom) has been included only to avoid kinetic terms with different signs (see also the new comment below (2.15)). The latter feature has been introduced by working with a unifying Lagrangian and it is not a general feature of the electric limit.
-
As stressed in our answer to point 1, in our work we do not modify the Clifford algebra. An alternative approach to Carroll fermions based on degenerate Carroll algebras, inducing a contraction of $so(1,d)$ has been considered in the literature and we mentioned it in footnote 1.
-
We mentioned in footnote 12 that the truncation defining the magnetic Carroll Lagrangian naively breaks parity, thus explaining the option to add mass terms with $\Gamma_\star$. We defer to further work an analysis of possible modifications of the action of parity on Carrollian spinors that might restore parity.

---

## Round 3 · Referee Report · Anonymous (Referee 4) · 2024-5-20

Report

The authors have given answers to the questions asked and have made the necessary changes. It is recommended for publication.

Recommendation

Publish (meets expectations and criteria for this Journal)

---

## Round 3 · Author Response

Dear Editor, dear Referees,

We thank the anonymous referee for reading our reply to their first report, and for suggesting new improvements of our manuscript in this second report. We added some further clarifications that we detail in the following list of changes. In particular, we tried to stress that the halving of degrees of freedom in the electric limit noted by the referee is an artifact induced by our way of rescaling the fields in eq. (2.13). One could as well rescale all spinors with the same factor of ˜c and obtain an electric action containing the same number of spinor components as in the original action. Our choice somehow stresses that in order to obtain a magnetic limit it is not sufficient to use the rescalings that we introduced in eq. (2.8) while discussing the Carrollian transformation rules of spinors. One crucially has to implement these rescalings in a `twisted' way as in (2.19), where we rescaled the two projected components in the opposite way in the two Dirac spinors entering the Lagrangian (2.10). Otherwise one gets an electric limit as detailed after (2.13).
We hope that these additional clarifications will make our paper suitable for publication.

---

## Round 3 · List of Changes

- Our electric and magnetic Carroll Lagrangian are invariant under the Carroll symmetries. In particular, the boost symmetry is obviously realised in the electric Lagrangian, however a non-trivial cancellation happens when checking that also the magnetic Lagrangian is boost invariant. Since our magnetic action in first-order formalism precisely matches the one written in the Lagrangian formalism, we expect that the same non-trivial cancellation happens when computing the Poisson brackets of its energy density. Since in our paper we mainly focus on the Lagrangian formalism, we decided to just comment about this below (2.40) rather than explicitly showing it.

- We slightly modified the text in the paragraph containing eq. (2.11) and we added eq. (2.12) to make the starting point of the following discussions more explicit. Consistently with the comments in our resubmission cover letter, we also stressed again above (2.13) that the rescalings proposed in that equation are not the only way to obtain an electric limit (the same rescalings were already mentioned at the beginning of the preceding paragraph, but we agree that it is more appropriate to stress the point in the paragraph dedicated to the Electric limit). We also added an explicit comment below (2.16) about the fact that even if we started from a magnetic'' rescaling of the relativistic fields we ended with electric'' Carrollian transformations thanks to the disappearing of ψ and χ in the limit. We also modified footnote 10 to explain that this is not to be considered as a general feature of the electric limit, but only as a consequence of our way to define a particular electric limit, eventually leading to the minimal electric Lagrangian (2.18).

- We modified the sentence about the unification of the limits below eq. (2.18) and we made more explicit that we provided a framework in which, differently from Dirac's action, we can perform either an electric or a magnetic limit. On the other hand, we are not able identify both limits at the same time as was done for bosonic theories via the c-expansion of the relativistic action in arXiv: 2110.02319. Motivated by the referee's comment we explained in footnote 12 at pag. 8 why the c-expansion of relativistic bosonic theories does not seem to provide a promising setup to unify the limits in the fermionic case. Indeed, in the bosonic case, to show that the electric (magnetic) case occurs in leading (subleading) order requires the application of a so-called Hubbard-Stratonovic transformation, which can only be performed if the leading divergence can be written as a square. This is true in bosonic actions where the kinetic term is a square, but this is no longer the case in fermionic actions. In fact, this difference with the bosonic case is precisely the reason that it took some effort to define a Carroll limit for fermions. Nevertheless, the c-expansion method applied to fermionic theories remains an interesting issue to explore.

- We added a small remark about the potential applications of Carrollian fermionic field theories in the study of flat space holography at the beginning of the last paragraph of pag. 2.

---

## Editorial Decision

published